# Crystal structures of the human elongation factor eEFSec suggest a non-canonical mechanism for selenocysteine incorporation

Malgorzata Dobosz-Bartoszek[1], Mark H. Pinkerton[2], Zbyszek Otwinowski[3], Srinivas Chakravarthy[4], Dieter Söll[5], Paul R. Copeland[2] & Miljan Simonović[1]

Selenocysteine is the only proteinogenic amino acid encoded by a recoded in-frame UGA codon that does not operate as the canonical *opal* stop codon. A specialized translation elongation factor, eEFSec in eukaryotes and SelB in prokaryotes, promotes selenocysteine incorporation into selenoproteins by a still poorly understood mechanism. Our structural and biochemical results reveal that four domains of human eEFSec fold into a chalice-like structure that has similar binding affinities for GDP, GTP and other guanine nucleotides. Surprisingly, unlike in eEF1A and EF-Tu, the guanine nucleotide exchange does not cause a major conformational change in domain 1 of eEFSec, but instead induces a swing of domain 4. We propose that eEFSec employs a non-canonical mechanism involving the distinct C-terminal domain 4 for the release of the selenocysteinyl-tRNA during decoding on the ribosome.

[1] Department of Biochemistry and Molecular Genetics, University of Illinois at Chicago, Chicago, Illinois 60607, USA. [2] Department of Biochemistry and Molecular Biology, Rutgers—Robert Wood Johnson Medical School, Piscataway, New Jersey 08854, USA. [3] Department of Biochemistry, University of Texas Southwestern Medical Center, Dallas, Texas 75390, USA. [4] Biophysics Collaborative Access Team/Illinois Institute of Technology, Sector 18ID, Advanced Photon Source, Chicago, Illinois 60439, USA. [5] Departments of Molecular Biophysics and Biochemistry, and Chemistry, Yale University, New Haven, Connecticut 06520, USA. Correspondence and requests for materials should be addressed to M.S. (email: msimon5@uic.edu).

Selenium is the only essential dietary micronutrient that is genetically encoded in all domains of life. It is found in proteins as the 21st amino acid selenocysteine (Sec). Mammals and humans, have 25 ubiquitously expressed seleno-proteins[1], many of which are essential. Selenoproteins and selenoenzymes are critical for the redox potential maintenance, protection of the genetic material and cell membrane from oxidative damage, regulation of the thyroid hormone metabolism, and control of gene expression and protein folding[2–4]. The replacement, either accidental or deliberate, of Sec with either serine (Ser) or cysteine (Cys) renders selenoenzymes either completely inactive or significantly catalytically impaired[5–7]. A tRNA$^{Sec}$ knockout mutant mouse is embryonically lethal[8] and mutations in enzymes facilitating selenoprotein synthesis cause systemic pathologies[3,4] including severe early-onset neurodegeneration[9–11]. This implies that the accurate decoding of the Sec codon and the correct placement of the Sec residue within the nascent selenoprotein chain is a fundamental biological process. However, the complex biosynthesis of eukaryotic selenoproteins is still poorly understood[4,12].

General translation elongation factors, eEF1A and EF-Tu, play a pivotal role in the elongation phase of protein synthesis. They are not only essential for the delivery of aminoacyl-tRNAs (aa-tRNAs) to the translating ribosome, but also for 'sensing' if the proper codon–anticodon interactions are established between the mRNA and the A-site aa-tRNA in the decoding centre of the small ribosomal subunit. The correct base pairing and interaction with the sarcin–ricin loop of the large ribosomal subunit stimulates the GTPase activity of EF-Tu and induces a major rearrangement of the protein factor structure. The ~90° rotation of domain 1 (D1) relative to domains 2 (D2) and 3 (D3)[13,14] disrupts the interactions between the aa-tRNA and the aminoacyl-recognition pocket, which leads to dissociation of EF-Tu:GDP from the aa-tRNA and the ribosome. Consequently, the acceptor stem of the aa-tRNA accommodates in the A-site on the large ribosomal subunit and its aminoacyl group becomes properly positioned within the peptidyl-transferase centre so that the reaction of peptide bond formation can occur[15,16]. Intriguingly, all aa-tRNAs are recognized and delivered to the ribosome by the same eEF1A and EF-Tu factors, except only for the Sec-tRNA$^{Sec}$, which is an obligate substrate for synthesis of selenoproteins[17,18]. Instead, the cotranslational insertion of Sec into a nascent selenoprotein is promoted by a specialized elongation factor, eEFSec in eukaryotes[19,20] and SelB in prokaryotes[21]. eEFSec is a translational GTPase that binds Sec-tRNA$^{Sec}$ with high affinity and stringent specificity, and plays a pivotal role during decoding[19] by delivering Sec-tRNA$^{Sec}$ to the site of translation in response to a particular in-frame UGA codon. The importance of eEFSec was first illustrated by the inability of the *EEFSEC* knockout mutant fruit flies to synthesize selenoproteins[22]. A hairpin structure in the selenoprotein mRNA, termed SElenoCysteine Insertion Sequence (SECIS), serves to differentiate the Sec UGA from the translational stop UGA codon. The bacterial SECIS element is located within the coding region immediately downstream of the Sec UGA[23], while the poorly conserved eukaryotic SECIS is located in the 3′-UTR of the mRNA[24]. It is presumed that SECIS anchors eEFSec and Sec-tRNA$^{Sec}$ near the ribosome, which could be important for avoiding the premature termination of translation. The Sec decoding process exhibits structural and functional variations. Whereas SelB binds directly to SECIS and promotes decoding unassisted, eEFSec can only do so in the presence of an exclusively eukaryotic auxiliary protein factor, SECIS Binding Protein 2 (SBP2)[25]. However, despite the divergent process idiosyncrasies across kingdoms, the core mechanism in which eEFSec (or SelB) plays the major role is conserved.

The current understanding of the mechanism of the Sec-tRNA specialized elongation factor is largely based on studies on prokaryotic model systems[21,26–33]. Prokaryotic SelB is composed of the N-terminal EF-Tu-like domain and an extended, structurally divergent, C-terminal domain 4 (D4)[29,33]. In contrast to EF-Tu, SelB has similar binding affinities for GTP and GDP[31], and neither eEFSec nor SelB require the guanine nucleotide-exchange factor activity to cycle between the GDP- and GTP-bound states. It is also suggested that the archaeal SelB does not undergo a large conformational change upon GTP-to-GDP transition[29], and that in mammals D4 can both modulate the GTPase activity and sense the nucleotide binding to eEFSec (ref. 34). These observations raised the question whether eEFSec, and by analogy SelB, promotes Sec incorporation by a mechanism distinct from the canonical mechanism based on EF-Tu. Herein, we determined the crystal structures of the intact human eEFSec in the GTP- and GDP-bound states. Also, by using a combination of site-directed mutagenesis, and *in vitro* binding and activity assays we identified and characterized functional sites responsible for Sec recognition, and GTP binding and hydrolysis. Our results show that the GTP-to-GDP exchange induces an unexpected conformational change in the C-terminal D4 of eEFSec and not in D1 as predicted by studies on EF-Tu and eEF1A. Although larger in its magnitude, this unexpected structural rearrangement in eEFSec resembles the domain dynamics observed in the universal translation initiation factor, IF2/eIF5B (ref. 35). We propose that eEFSec and SelB employ a non-canonical mechanism for Sec-tRNA$^{Sec}$ release, which is pivotal for the read-through of the Sec UGA codon.

## Results

**Overall structure of human eEFSec.** As the first step towards addressing the mechanism of decoding of the Sec UGA codon in higher organisms, we determined the crystal structures of the intact human eEFSec in complex with GDP and non-hydrolyz-able GTP analogues, GDPNP and GDPCP. Our results show that four domains of human eEFSec fold into a chalice-like structure resembling the archaeal SelB (ref. 29) and IF2/eIF5B (ref. 35).

The isomorphous orthorhombic crystals of eEFSec:GDPNP and eEFSec:GDPCP diffracted X-rays to 3.4 and 2.7 Å, respectively, while the monoclinic crystals of eEFSec:GDP diffracted X-rays to 3.3 Å (Table 1). With the exception of flexible loops (residues 32–42, 70–80, 192–195, 383–403, 435–438, 524–526 and 544–569), the entire protein backbone was traced (Fig. 1, Supplementary Fig. 1). The final models were refined to $R_{work}/R_{free}$ of 0.24/0.29 (eEFSec:GDPNP), 0.24/0.29 (eEFSec:GDPCP) and 0.30/0.34 (eEFSec:GDP) (Table 1). The asymmetric unit in each crystal form contained either a head-to-head (eEFSec:GDPCP, eEFSec:GDPNP) or a head-to-tail dimer (eEFSec:GDP). Given that monomers in each dimer are nearly indistinguishable and that eEFSec is an obligate monomer in solution, the crystallographic dimers are most likely of no physiological significance. Human eEFSec adopts a chalice-like structure composed of four domains: the N-terminal D1 (residues 1–215), D2 (residues 224–304), D3 (residues 310–455), and the C-terminal D4 (residues 477–575) (Fig. 1a). Domains 1–3 represent the cup, the linker region (residues 469–476 and 576–582) is the stem, and D4 is the base of the chalice (Fig. 1b). The height of the chalice is ~100 Å, while its width is variable; it is the largest at the cup (60 Å), significantly smaller at the foot (30 Å), and the smallest at the stem (20 Å). The N-terminal D1–3 folds into an EF-Tu-like structure (see below) harbouring the GTPase site and the putative Sec-binding pocket. The 6-stranded β-sheet of D1 is enclosed by 7 α-helices. An 8-residue long loop connects D1 with a β-barrel structure of D2, which is composed

**Table 1 | Data collection and refinement statistics.**

|  | eEFSec:GDPNP* | eEFSec:GDPCP† | eEFSec:GDP |
|---|---|---|---|
| *Data collection* |  |  |  |
| Space group | C 2 2 2₁ | C 2 2 2₁ | P 2₁ |
| Cell dimensions |  |  |  |
| a, b, c (Å) | 92.3, 112.4, 327.7 | 94.3, 113.2, 329.7 | 58.7, 96.9, 125.4 |
|  |  |  | β = 90.25° |
| Resolution (Å)‡ | 46.00–3.38 (3.50–3.38) | 49.00–2.72 (2.81–2.72) | 43.00–3.00 (3.05–3.00) |
| $R_{sym}$ or $R_{merge}$ | 0.09 | 0.23 | 0.13 |
| $I/\sigma I$‡ | 19.7 (1.2) | 14.5 (1.0) | 7.6 (0.4) |
| Completeness (%)‡ | 98.5 (88.9) | 100 (100) | 92.4 (51.8) |
| Redundancy‡ | 9.5 (6.1) | 19.4 (12.1) | 3.7 (2.6) |
|  |  |  |  |
| *Refinement* |  |  |  |
| Resolution (Å) | 46.00–3.40 | 49.00–2.72 | 43.00–3.25 |
| No. of reflections | 21,650 | 46,503 | 22,047 |
| $R_{work}/R_{free}$ | 0.24/0.29 | 0.24/0.29 | 0.30/0.34 |
| No. of atoms |  |  |  |
| Protein | 6,604 | 7,259 | 5,983 |
| Ligand/ion | 66 | 66 | 56 |
| Water |  | 32 | 21 |
| B-factors |  |  |  |
| Protein | 32.5 | 79.4 | 81.6 |
| Ligand/ion | 38.2 | 76.4 | 107.4 |
| Water |  | 77.0 | 68.2 |
| r.m.s. deviations |  |  |  |
| Bond lengths (Å) | 0.010 | 0.014 | 0.009 |
| Bond angles (°) | 2.04 | 1.48 | 1.96 |

*Two crystals were used in data collection.
†Four crystals were used in data collection.
‡Values in parentheses are for highest-resolution shell.

of 8 antiparallel β-strands and a flanking, short α-helix. D3 harbours 7 antiparallel β-strands and continues into the linker region *via* a long helix α8. The linker is composed of two β-strands; the first strand, β23, arises from D3 and continues into D4, which folds into a small β-barrel flanked by a pair of α-helices. The second strand of the linker region, β28, runs antiparallel to β23 and ends with an α-helical turn (residues 583–588) that sits below D3. The most C-terminal segment (residues 589–595) folds back below D3 and almost perpendicularly to the plane of the linker region while adopting a β-turn structure.

**eEFSec is a structural chimera of EF-Tu and IF2/eIF5B.** A detailed comparison with a myriad of translation GTPases acting on the ribosome reveals that human eEFSec is a structural chimera of the general translation elongation (EF-Tu/eEF1A) and initiation (IF2/eIF5B) protein factors[29]. Also, eEFSec resembles more closely the archaeal rather than the bacterial SelB.

The N-terminal domain of human eEFSec resembles EF-Tu. Several structural differences that could be of functional significance have been noted. The overlay of D1 (r.m.s.d. of 1.5 Å) shows that EF-Tu harbours two well-ordered α-helical insertions that sit atop the GTPase site. By contrast, these regions are shorter in eEFSec (residues 32–42 and 186–202) and partially disordered in our crystals. Likewise, the dorsal side of eEFSec harbours a partially disordered insertion (residues 57–87) where EF-Tu contains a well-ordered loop. Further, D2 from eEFSec and EF-Tu are similar (r.m.s.d. of 1.5 Å) with the only difference present in loop β10-β11, which is significantly shorter in eEFSec. Lastly, the overlay of D3 (r.m.s.d. of 1.7 Å) revealed that eEFSec contains insertions in several solvent-exposed loops: loop β17–β18 (residues 352–373), located at the dorsal face of eEFSec, β21–β22 (residues 432–444) at the interface of D1 and D3, and β18–β19 (residues 378–410).

The structural homology with the bacterial SelB is restricted to the EF-Tu-like domain, while it extends to D4 in case of the archaeal SelB (Supplementary Fig. 2a,b). The global super-impositioning of the archaeal (356 residues) and bacterial SelB (323 residues) onto human eEFSec yields r.m.s.d. values of 2.2 and 2.1 Å, respectively. The overlay of D1 only results in much lower values ranging between 1.1 and 1.4 Å. The main differences in D1 are within switch 1 and around the GTPase site where several enlarged loops in eEFSec are partially disordered (that is, residues 59–85 and loop β6–α6). An analogous comparison of D2 yields values of 1.2 Å (archaeal SelB) and 1.3 Å (bacterial SelB); the main difference there is in the orientation of loop β7–β8 (residues 233–238). Similarly low r.m.s.d. values of 1.3–1.6 Å were obtained when atoms from D3 were used in calculation only. The only discrepancy is in loops β17–β18, β18–β19, and β21–β22, which are enlarged in eEFSec.

The most marked differences are present in the C-terminal D4. The bacterial D4 consists of four winged-helix folds and is rotated ∼90° around the linker region when compared to the archaeal SelB and human eEFSec (Supplementary Fig. 2b). Although conservation of the archaeal and human D4 is not strict, as the human enzyme harbours an additional α-helix and a longer C-terminal segment (Supplementary Fig. 2a), closer inspection shows that the orientation of human D4 is stabilized by interactions between loop β28-α11 at the extreme C-terminus and residues in D3 (Supplementary Fig. 3a). In particular, an H-bond is formed between the conserved Glu372 from D3 and Lys582 from the C-terminal segment and it is present in both the GDP- and GDPCP-bound structures (Supplementary Fig. 3a,b). The Glu-Lys pair is found in the archaeal SelB (for example, Glu325, Lys388), but not in the bacterial orthologue (Supplementary Fig. 4). This led us to hypothesize that interactions within this 'hinge' region could be of importance for the interdomain interactions and perhaps domain orientation.

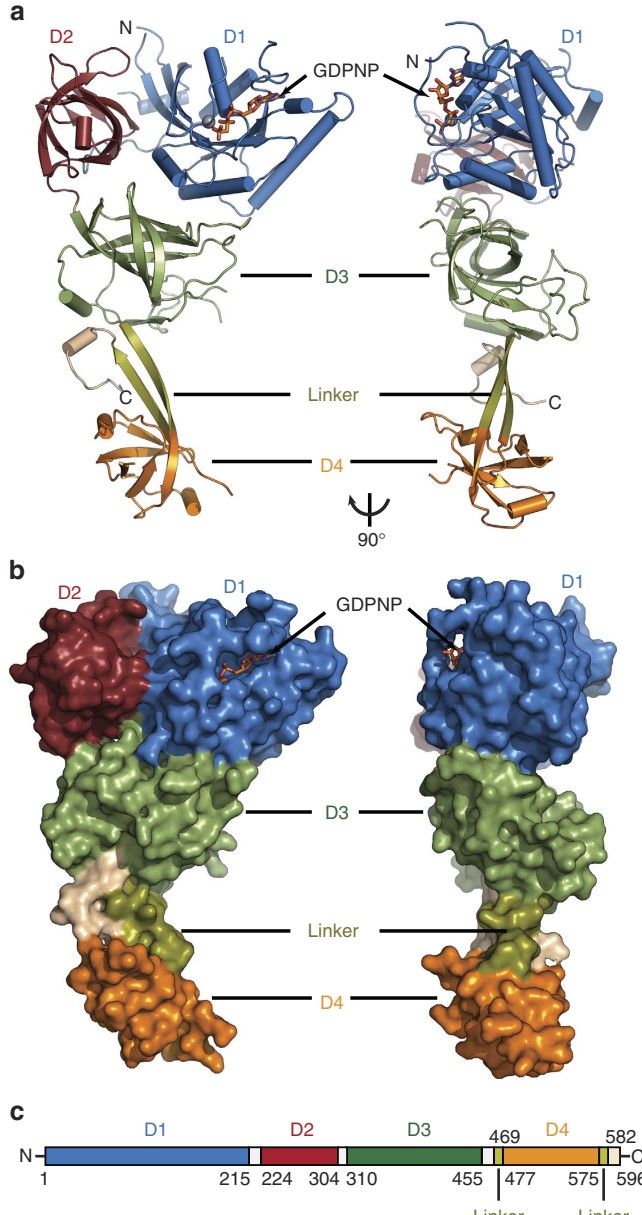

**Figure 1 | The overall structure and domain organization of human eEFSec.** (**a**) Cartoon and (**b**) surface representation diagrams of the chalice-like structure of human eEFSec shown in two views rotated ∼90° clockwise around vertical axis. Individual domains, linker and extreme C-terminus regions are labelled and coloured according to the scheme in (**a**). GDPCP is shown as sticks and $Mg^{2+}$ as a grey sphere. (**c**) Schematic diagram showing domain organization in human eEFSec. White bars denote regions connecting individual domains.

We assessed the importance of the 'hinge' region by mutational and activity studies. The K582A and $^{582}$KRYVF$^{586}$ -> AAAAA variants express at a lower level and are less stable when compared to the WT eEFSec. On the other hand, the $^{583}$RY$^{584}$ -> AA double substitution does not alter the structure and activity of eEFSec in vitro (Supplementary Fig. 3c). Our results suggest that the salt bridge between Gly372 and Lys582 is structurally important, but additional experiments are needed to assess its functional role.

Previous phylogenetic studies proposed that eEFSec and SelB are in a closer evolutionary relationship with IF2/eIF5B than with EF-Tu[36]. This notion is supported by a relatively good structural

agreement between individual domains of human eEFSec, archaeal IF2[35] (r.m.s.d. of 2.7, 1.6 and 2.8 Å for D1, D2 and D4, respectively), and yeast eIF5B[37] (r.m.s.d. of 3.0, 1.7 and 3.1 Å for D1, D2 and D4, respectively). In spite of the divergence within D3, which was omitted from calculations, the overall domain organization and the shape of human eEFSec and IF2/eIF5B are similar (Supplementary Fig. 2c). Hence, we conclude that eEFSec, just like SelB, is a structural chimera of EF-Tu and IF2/eIF5B.

**GTP-to-GDP exchange induces a conformational change in D4.** Studies on the bacterial protein synthesis established that a conformational change in EF-Tu coupled to GTP hydrolysis is critical for the elongation phase of translation. Only in the GTP-bound state EF-Tu binds and delivers aa-tRNAs to the translating ribosome. After the anticodon–codon interactions are formed, the GTPase activity and the conformational changes in EF-Tu are induced. In particular, D1 rotates away from D2 and D3, and the globular structure of the GTP-bound state is transformed into a more extended structure of the GDP-bound EF-Tu[14,38]. The disruption of the aminoacyl-binding pocket causes EF-Tu:GDP to dissociate from aa-tRNA and the ribosome[14]. The CCA-end is then accommodated within the peptidyl-transferase centre and the aminoacyl group is poised for the reaction of peptide bond formation. It was recently suggested that the analogous mechanism is employed by the mammalian eEF1A (ref. 39). The question, however, remained if the Sec-tRNA specific elongation factors undergo the same conformational change on the GTP-to-GDP exchange as EF-Tu and eEF1A. The study on the archaeal SelB suggested otherwise[29], but the results were questioned[31] because the functional states were captured by ligand soaking and not by co-crystallization. To address these fundamental questions, we performed a detailed analysis of the GDP- and GTP-bound states of human eEFSec.

Remarkably, the structures of the GTP- and GDP-bound states of human eEFSec are very similar (Fig. 2a). Superimposing of eEFSec:GDP onto eEFSec:GDPNP and eEFSec:GDPCP yields r.m.s.d. values of 2.5 and 2.1 Å, respectively. The most surprising observation is that D1 assumes a similar orientation relative to D2 and D3 in both functional states of eEFSec. Even more unexpected was the finding that the GTP-to-GDP exchange induces ∼26° swing of the C-terminal domain D4 away from the predicted tRNA-binding site (Fig. 2a), which results in >15 Å translation of the entire domain. D1 and D2 ratchet slightly in the opposite directions: D1 moves towards the ventral side (that is, the tRNA-binding face) of the molecule, while D2 rotates the other way. The domain movements, which are most noticeable when the molecule is viewed from the side (Fig. 2a, Supplementary Movie 1), cause the GTPase site to relax and the Sec-binding pocket to constrict. This is not entirely surprising since one would expect that the removal of the γ-phosphate would lead to the opening of the nucleotide-binding site. Also, the tightening of the Sec-binding pocket would presumably cause a sufficient decrease in the binding affinity towards the Sec moiety, which would lead to Sec-tRNA$^{Sec}$ release. The movement of D4, however, seems peculiar. Given that it was suggested that D4 might 'sense' the nucleotide binding and regulate the GTPase activity of the mammalian eEFSec (ref. 34), it is plausible that the domain movement may indeed be important. Similar structural rearrangements have been reported for the archaeal SelB (ref. 29) and the translation initiation factor IF2/eIF5B (refs 35,40) (Supplementary Fig. 2d). Whereas the domain movements in IF2/eIF5B are considered as functionally important, the analogous rearrangements in SelB were not discussed outside the realm of local structure dynamics. We wondered whether a reasonable structural model of the 'canonical' GDP-bound state of eEFSec, in

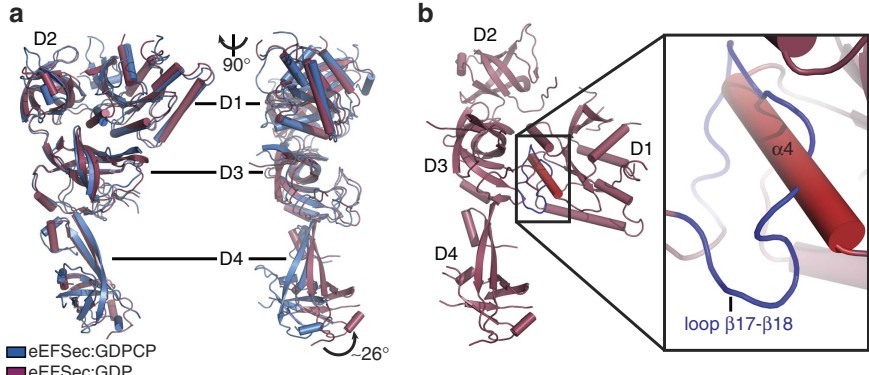

**Figure 2 | The GTP-to-GDP exchange on human eEFSec induces a conformational change in D4 and not in D1.** (**a**) The global superimpositioning of eEFSec:GDPCP (blue) and eEFSec:GDP (light red) reveals a lack of the canonical conformational change in the EF-Tu-like domain. Instead, the C-terminal D4 swings ∼26° towards the dorsal face of the molecule and away from the tRNA-binding site. Two views related by ∼90° clockwise rotation around a vertical axis are shown. The view on the left is oriented so that the tRNA-binding site is in the paper plane. The view on the right is oriented so that the tRNA-binding site (or ventral face) is on the left and perpendicular to the paper plane. (**b**) Modelling of the canonical conformational change in eEFSec reveals that the movement of D1 might be prevented by a steric clash between the enlarged loop β17–β18 (blue) in D3 and helix α4 (red) in D1. The steric clash is highlighted with black box and shown in a close-up view.

which D1 adopts the same orientation as in the GDP-bound EF-Tu, could be designed. Based on our sequence analysis and structural modelling we propose that loop β17–β18 in D3, which is absent from the canonical elongation factors and relatively conserved in eEFSec and SelB, might hinder rotation of D1 in eEFSec (Fig. 2b). This loop is on the dorsal side (that is, opposite from the tRNA-binding face) of eEFSec and it would clash with helix α4 if the canonical conformational change were possible. Therefore, we suggest that the evolutionary pressure yielded structural elements in eEFSec and SelB that would inhibit rotation of D1 upon GTP hydrolysis, which, in turn, would lead to a distinct mechanism for Sec-tRNA[Sec] release.

The question, however, could be raised whether the crystal packing in the eEFSec:GDP complex crystals somehow hindered the movement of D1 or whether the D4 is structurally flexible, which would undermine the significance of the observed structural differences. To assess these possibilities, we analysed eEFSec:GDP and eEFSec:GDPCP by size-exclusion chromatography coupled to Small-Angle X-ray Scattering (SEC-SAXS). Our results show that eEFSec adopts a strikingly similar molecular shape irrespective of the bound nucleotide ($R_g$ of ∼35 Å). Importantly, the crystal structures agree with the corresponding SAXS-derived molecular envelopes quite well (Supplementary Fig. 5a,b), while the predicted 'canonical' model of eEFSec:GDP could not be superimposed onto the eEFSec:GDP envelope (Supplementary Fig. 5c). Remarkably, a side-to-side comparison of SAXS envelopes, which are of significantly lower resolution than the X-ray crystal structures, clearly shows that D4 of eEFSec adopts different orientations in the GDP- and GTP-bound states (right panels, Supplementary Fig. 5a,b). We conclude that the conformational change between the GDP and GTP states observed in eEFSec in crystals also occurs in solution. Thus, the conformational change in the GDP-bound state of eEFSec is not a crystallization artifact. We also argue that the eEFSec complex structures presented herein, though obtained in the absence of the decoding complex, represent the physiologically relevant states. This would be consistent with the fact that EF-Tu adopts the same structure in isolation[13,14,41] as in complex with aa-tRNA[38,42,43] and ribosome[15,16]. Given that our results agree with the earlier findings by Ban and co-workers with archaeal SelB (ref. 29), we propose that eEFSec and its orthologs utilize a non-canonical mechanism for the release of Sec-tRNA[Sec] during decoding. This particular mechanism requires minor

adjustments in D1 and D2, and the major conformational change in D4. Such a mechanism perhaps arose due to specific and distinct requirements of the Sec-decoding machinery that are not present in the canonical system.

**Changes in functional sites upon nucleotide exchange.** The nucleotide exchange induces a series of small, but important structural adjustments in functional sites of eEFSec. As we have mentioned, the GDP binding causes slight ratchet of D1 and D2. As a consequence, the GTPase site relaxes and the Sec-binding pocket constricts.

The GTPase site is located in D1 and it is composed of conserved elements found in other small GTPases: the P loop ([14]GxxxxGKT[21]), switch 1 (residues 32–47), switch 2 ([92]DxxGH[96]), the guanine-binding sequence ([146]NKxD[149]) and a divalent metal ion (Fig. 3a). Apart from partially disordered switch 1, the GTPase site is well ordered in both complex crystals. The guanine ring of the analogue is stacked between the aliphatic chain of Lys147 and loop 190 ([188]PGGP[191]). The Watson–Crick face interacts with the side chain of Asp149, while the Hoogsteen side forms H-bonds with the backbone amide of Ala186 and the side chain of Asn146 (Fig. 3a). These interactions are preserved in the GDP-bound complex (Fig. 3b). The ribose hydroxyls of GDPCP are within H-bonding distance from the side-chain carboxyl of Glu44. The backbone amides of the P loop ([19]GKTA[22]) interact with non-bridging oxygens of α- and β-phosphates. The β- and γ-phosphates are held in place by interactions with $Mg^{2+}$ and the backbone amide of Gly95 in switch 2. The side chains of Thr21 and Thr48 in switch 1, and two water molecules, W1 and W2, complete the coordination of $Mg^{2+}$ (Fig. 3a and Supplementary Fig. 1a). W2 is oriented for interaction with $Mg^{2+}$ by H-bonds with Asp92 from switch 2. Perhaps more importantly, the side chain of the presumed catalytic His96 points away from the γ-phosphate, implying that the structure of the GTPase site of eEFSec in isolation does not adopt active conformation and cannot catalyse the GTP hydrolysis. This finding is not unprecedented, since EF-Tu and EF-G adopt the active conformation when bound to the ribosome. In both instances, the catalytic His87 is repositioned towards the GTP γ-phosphate through an H-bond with A2662 of the sarcin–ricin loop of the 23S ribosomal RNA[16,44,45]. It is, therefore, reasonable to suggest that a similar interaction

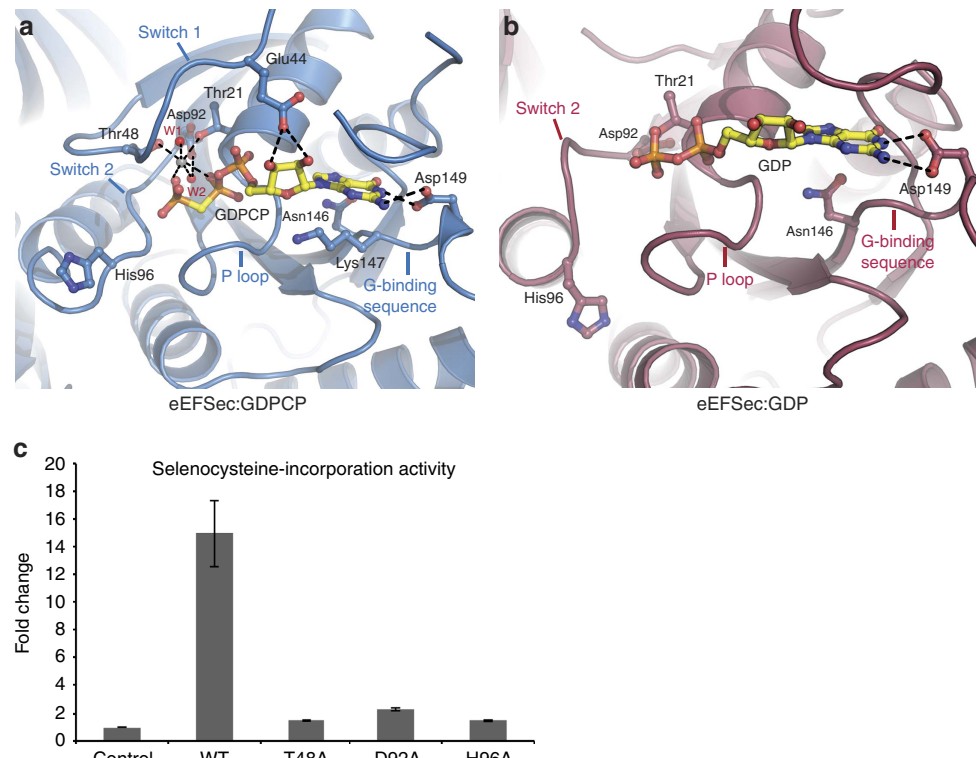

**Figure 3 | The structure of the GTPase site in human eEFSec.** (**a**) The GTPase site is located in D1 and is composed of the structural elements conserved in small GTPases: the P loop, switch 1, switch 2, the guanine-binding sequence and $Mg^{2+}$ ion. The binding of the GTP analog, GDPCP, causes partial ordering of switch 1, which orients Thr48 and Asp92 to interact with $Mg^{2+}$ and waters 1 and 2 (W1, W2). (**b**) The GTP-to-GDP transition induces structural rearrangements mainly restricted to switch 1 and switch 2 regions. In the GDP-bound state, switch 1 is almost completely disordered and switch 2 adopts a different orientation. The conserved residues in the GTPase site are shown as sticks, water molecules are shown as red spheres, $Mg^{2+}$ is a grey sphere and H-bonds are dashed lines. (**c**) Mutations of conserved residues in GTPase site (Thr48, Asp92 and His96) severely impair read-through of the Sec UGA codon and selenoprotein synthesis *in vitro*. Error bars represent standard deviation (s.d.) calculated from three replicates.

with the ribosome stimulates the GTPase activity of eEFSec and SelB as well.

The binding of GDP to eEFSec causes disorder in switch 1 and slight repositioning of switch 2 further away from the nucleotide (Fig. 3b and Supplementary Fig. 1b), which yields a more open conformation of the GTPase site in the GDP-bound state. In particular, a segment of switch 1 that harbours Thr48 is ordered in the GTP-bound state as it provides the lid over the pocket accommodating the γ-phosphate (Fig. 3a). By contrast, this segment is disordered in the GDP complex (Fig. 3b). In addition, loop β3–α3 and helix α3 of switch 2 tilt ∼5 Å closer to D2 and the Sec-binding pocket, and away from the GTPase site. As a result, Asp92 and His96 point away from the GTPase site and are not properly oriented for $Mg^{2+}$ coordination and GTP binding, respectively. This explains why $Mg^{2+}$ is not present in eEFSec:GDP even though the sample contained 5 mM $MgCl_2$ during purification and crystallization trials. Our results are consistent with the crystal structure of the GDP-bound state of rabbit eEF1A2 (ref. 39).

Given their role in GTP/GDP binding to EF-Tu, we decided to assess if Thr48, Asp92 and His96 are important for eEFSec function. We found that although T48A, D92A and H96A variants bind GTP and GDP with high affinity (Table 2 and Supplementary Fig. 6), they cannot promote the UGA codon read-through and selenoprotein synthesis *in vitro* (Fig. 3c). Since these mutations do not compromise the structure of eEFSec and GTP/GDP binding (that is, all mutant constructs purified as monomeric proteins), we conclude that Thr48, Asp92 and His96 are important for GTP hydrolysis. Further structural and

enzymatic studies utilizing a complete human decoding complex are needed to define roles of Thr48, Asp92 and His96 in this process. This is particularly true for His96, the amino acid that was suggested to be essential for the ribosome-induced GTPase activity of EF-Tu. His96 is positioned relatively far from GTP analogues and GDP in our structures, which implies that this particular segment of the GTPase site may undergo an additional conformational change when bound to the translating ribosome.

Further, the GTP-to-GDP exchange causes structural rearrangements in the Sec-binding pocket. This pocket is located at the interface of D1 and D2 and it is composed of Phe53 from D1 and Asp229, His230 and Arg285 from D2 (Fig. 4a,b and Supplementary Fig. 4). Asp229 and Arg285 are conserved across all kingdoms, while Phe53 and His230 are replaced with Tyr and Arg in the bacterial SelB. A segment of strand β7 and the entire β7-β8 turn, which are downstream of Asp229 and His230, are completely disordered in the GDP-bound structure. Further, turn β10–β11 (residues 258–264), loop β14–β15, and helix α7 (residues 290–300), which form the mouth of the pocket move ∼3.5 Å towards D1. This might explain why the GTP-bound state of mammalian eEFSec and bacterial SelB exhibits a markedly stronger binding affinity towards Sec-tRNA^Sec (ref. 30). Lastly, on the dorsal side of D3, loops β15–β16 (residues 321–331) and β18–β19 (residues 365–411) swing in the same direction as D4 albeit to a lesser extent. Although small in magnitude, the movement of D3 during nucleotide exchange could contribute to the release of Sec-tRNA^Sec. The question remains as to how eEFSec (and SelB) select Sec-tRNA^Sec over all other aa-tRNAs? It was thought that the positive charge in the binding pocket

**Table 2 | Binding of guanine nucleotides and analogues to WT and variants of eEFSec.**

| Ligand | Protein | $K_d$ (μM) | ΔH (kcal mol$^{-1}$) | ΔS (cal mol$^{-1}$ per deg) |
|---|---|---|---|---|
| GDP | WT | 0.19 ± 0.05 | −17.19 ± 0.30 | −26.90 |
| | T48A | 0.24 ± 0.05 | −16.11 ± 0.25 | −23.70 |
| | D92A | 1.50 ± 0.13 | −9.41 ± 0.13 | −4.90 |
| | H96A | 1.30 ± 0.24 | −18.14 ± 0.45 | −33.90 |
| GTP | WT | 1.21 ± 0.26 | −20.57 ± 0.67 | −41.90 |
| | T48A | 1.82 ± 0.11 | −17.41 ± 0.18 | −32.10 |
| | D92A | 8.47 ± 0.49 | −9.88 ± 0.49 | −9.92 |
| | H96A | 1.60 ± 0.75 | −19.06 ± 1.37 | −37.40 |
| GDPNP | WT | 1.20 ± 0.47 | −16.05 ± 0.77 | −26.70 |
| GDPCP | WT | 1.85 ± 0.24 | −20.23 ± 0.46 | −41.60 |
| GTPγS | WT | 0.48 ± 0.06 | −16.87 ± 0.19 | −27.70 |

$K_d$, dissociation constant; ΔH, enthalpy change; ΔS, entropy change.
Human eEFSec contains a single nucleotide-binding site; N = 1.

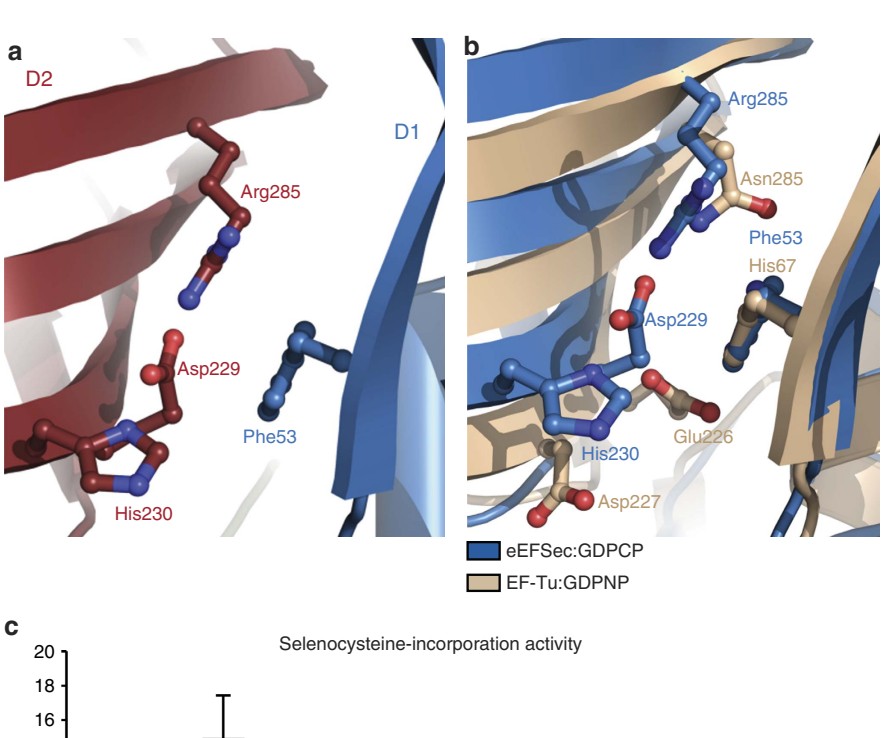

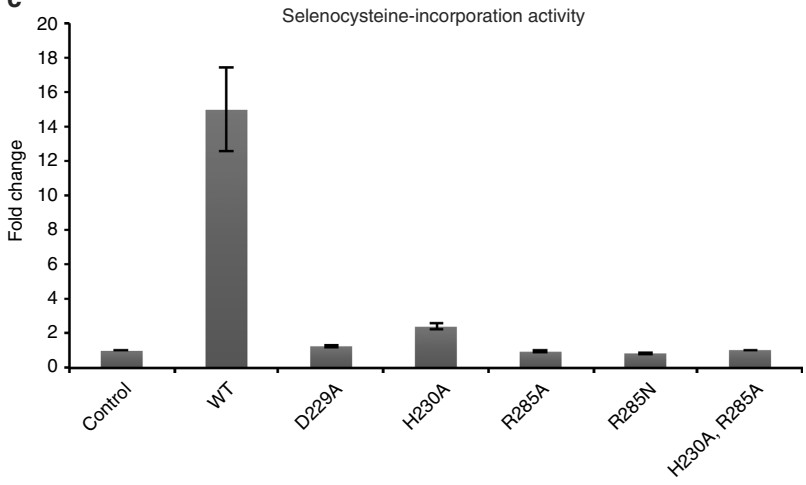

**Figure 4 | The structure of the putative Sec-binding pocket in human eEFSec.** (**a**) The site is located at the interface of D1 (Phe53; blue) and D2 (Asp229, His230, Arg285; red). Asp229 and Arg285 are conserved across the kingdoms, whereas His230 and Phe53 are present in eukaryotic and archaeal orthologs, but are replaced by Arg and Tyr in bacterial SelB. (**b**) The superimpositioning of the amino acid-binding sites reveals differences between eEFSec (blue) and EF-Tu (beige). The overall positive charge of the site in eEFSec is suggested to complement for negatively charged Sec moiety of Sec-tRNA$^{Sec}$. (**c**) The replacement with alanine of any of the residues in the Sec-binding pocket (Asp229, His230, and Arg285) completely abolishes the read-through of Sec UGA codon and selenoprotein synthesis *in vitro*. Error bars represent s.d. calculated from three replicates.

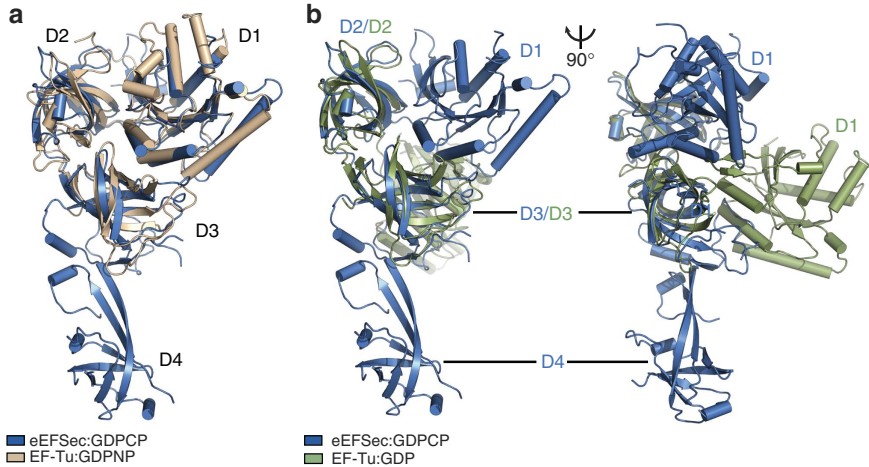

**Figure 5 | GTP analogues lock eEFSec in the GTP-bound state.** The overlay of human eEFSec:GDPCP (blue) onto EF-Tu:GDPNP (**a**, beige) and EF-Tu:GDP (**b**, green), reveals that the arrangement of D1-3 in eEFSec:GDPCP is similar to that in the GTP-bound state of EF-Tu. Thus, GDPCP and GDPNP are optimal GTP mimics when bound to human eEFSec. The orientation of D1 in EF-Tu:GDP can only be seen when the view in (**a**) and (**b**, left) is rotated ~90° clockwise around the vertical axis.

complements the negative charge of the Sec moiety[29]. We tested if Asp229, His230 and Arg285 are functionally significant by engineering D229A, H230A, R285A, R285N and H230A-R285A variants. The R285N mutation was introduced with the aim to mimic the aminoacyl recognition site of EF-Tu. In contrast to the archaeal SelB that requires at least one positive charge for its activity[29], each single residue substitution obliterated the ability of eEFSec to promote Sec incorporation (Fig. 4c). Due to the relatively large distance from the GTPase site, we conclude that all these substitutions most likely affect the binding of Sec-tRNA$^{Sec}$, rather than the GDP/GTP binding or the GTPase activity of eEFSec. However, to fully understand significance of the Sec-binding pocket and its role in the aminoacyl group recognition, additional studies involving WT and mutant eEFSec constructs, and Sec-tRNA$^{Sec}$ are warranted.

Lastly, it is important to mention significant corollaries derived from our studies. The question was raised whether GDPNP is a faithful mimic of GTP when bound to Sec elongation factors[31]. Our results unambiguously show that the GTPase site of eEFSec is the same when bound to GDPCP and GDPNP, and that both analogues trap eEFSec in the GTP-bound state (Fig. 5 and Supplementary Fig. 7). Consistent with our structural results, eEFSec binds GTP, GDPCP and GDPNP with similar affinities, while the binding affinity for GTPγS is similar to that observed for GDP (Table 2 and Supplementary Fig. 6); the discrepancy among analogues is likely due to spontaneous hydrolysis of GTPγS. In addition, unlike EF-Tu[46], eEFSec cannot discriminate GDP and GTP (Table 2). Taken together, our data suggest that GDPNP and GDPCP are equally good analogues of GTP when bound to eEFSec and by extension to SelB, and that previous structural results on the archaeal SelB are valid[29]. However, translational GTPases seem to have different responses to GTP analogues when bound to the ribosome. Namely, GDPNP failed to induce proper positioning of the catalytic His in RF3[47] and EF-G[48]. By contrast, GDPCP allowed EF-Tu and EF-G to adopt the active conformation on the ribosome[16,44,45,49,50]. Thus, it remains to be seen if both GTP analogues allow eEFSec and SelB to adopt the active conformation when in complex with the ribosome.

**The mechanism of decoding of the Sec UGA codon.** How does eEFSec facilitate the elongation phase of the selenoprotein mRNA translation? Our results demonstrate that the nature of

conformational changes in eEFSec is different from that in the general elongation factors, eEF1A and EF-Tu. Consistent with the structural results presented herein, we propose that eEFSec (and SelB) employs a non-canonical mechanism for the cognate tRNA release during decoding of the Sec codon (Fig. 6). The GTP-bound state of eEFSec is capable of recognizing and binding Sec-tRNA$^{Sec}$. The binding might involve interactions between D4 of eEFSec and the acceptor-TΨC elbow and the variable arm of tRNA$^{Sec}$. The ternary eEFSec:GTP:Sec-tRNA$^{Sec}$ complex is then tethered near the ribosome by the SBP2–SECIS complex. The mechanism governing interaction between these two complexes is not clear and further structural studies are needed. When ribosome reaches the Sec UGA codon, eEFSec delivers Sec-tRNA$^{Sec}$ to the A-site. D4 of eEFSec most likely points in the direction of the central protuberance with its $^{549}$KKRAR$^{553}$ sequence[34] and poised to interact with SBP2. Formation of the codon–anticodon interactions and interaction with the ribosomal RNA stimulates the GTPase activity of eEFSec in a mechanism analogous to EF-Tu. GTP hydrolysis induces a slight ratchet of D1 and D2, which results in opening of the GTPase site and constriction of the Sec-binding pocket (Fig. 6). These movements contribute to the decrease of the binding affinity of eEFSec towards the Sec moiety of Sec-tRNA$^{Sec}$. Concurrently, the lever-like movement of D4 towards the dorsal side of eEFSec leads to the release of Sec-tRNA$^{Sec}$, and dissociation of eEFSec from SBP2–SECIS and the ribosome (Fig. 6). This particular mechanism is applicable to SelB with a significant distinction that the prokaryotic process does not involve SBP2. It could be that D4 of the prokaryotic SelB interacts more closely with SECIS and/or ribosomal proteins rather than eEFSec. Moreover, it is plausible that D4 in eEFSec/SelB undergoes an additional rearrangement once bound to the ribosome and the A-site Sec-tRNA$^{Sec}$, analogous to IF2/eIF5B (Supplementary Fig. 2d). Namely, D4 of IF2/eIF5B translates ~4 Å upon nucleotide exchange, but then undergoes, together with D3 and linker, a much larger rotation (~50°) after interacting with the acceptor arm of the initiator tRNA and the ribosome (Supplementary Fig. 2d). The large domain movement and distortion of the tRNA body were shown to be of functional importance[40]. In any instance, the unexpected domain motions coupled to GTP hydrolysis observed in eEFSec are likely to be conserved across species. But, what is the rationale for having a distinct decoding mechanism for Sec? The ability of selenium to reversibly react

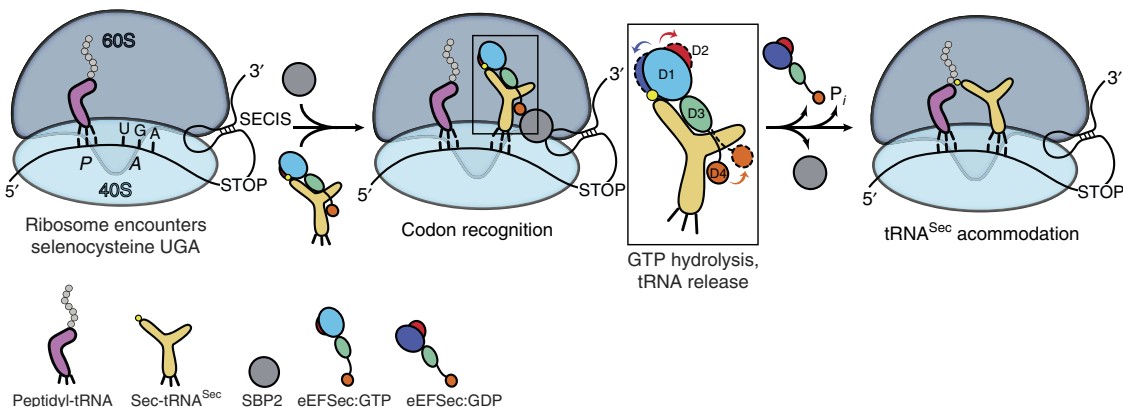

**Figure 6 | A model of decoding of the Sec UGA codon by human eEFSec.** The 80S ribosome pauses when encountering the UGA codon. The ternary eEFSec:GTP:Sec-tRNA$^{Sec}$ complex and SBP2 (grey sphere) bind to the ribosome through interactions with the SECIS element, which is in the 3′UTR of the selenoprotein mRNA. After codon recognition, eEFSec hydrolyses GTP and D1 (shades of blue) and D2 (red) move in a ratchet-like motion towards and away from the tRNA-binding (or ventral) side, respectively. Also, the domain D4 (orange) swings ∼26° away from the ventral side. These movements are emphasized in the boxed inset; the GTP- and GDP-bound states of eEFSec are shown in solid and dashed lines, respectively. After eEFSec:GDP dissociation, Sec-tRNA$^{Sec}$ accommodates and formation of the peptide bond occurs. Note: the variable arm of tRNA$^{Sec}$ (pointing to the right), D4 of eEFSec, SBP2 and SECIS are most likely oriented perpendicularly to the plane of the paper and towards the reader.

with oxygen and reactive oxygen species is of critical importance for organisms living in oxygenated atmosphere since it provides the basis for the significantly higher efficiency of enzymes containing Sec over those harbouring Cys (ref. 51). Consequently, the evolutionary pressure to have the Sec residue precisely incorporated into the protein chain yielded a complex decoding process divergent from the regular mechanism based on eEF1A and EF-Tu. The recent intriguing discovery that codons other than UGA, both nonsense and sense, encode for Sec in various microorganisms[52] ensures more surprises about Sec incorporation to be revealed.

## Methods

**Purification of human eEFSec.** Human eEFSec gene was cloned into pET15b containing N-terminal 6xHis tag. The plasmid was transformed into *E. coli* Lemo21(DE3) and the cells were grown at +37 °C in Luria–Bertani (LB) medium, supplemented with ampicillin, chloramphenicol, and 0.2 mM L-rhamnose. Once OD measured at 600 nm reached ∼0.6 units, the overnight expression at +16 °C was induced with 0.4 mM IPTG. The cells were harvested by centrifugation and resuspended in the lysis buffer (50 mM HEPES pH 7.5, 300 mM NaCl, 10% gly-cerol, 5 mM β-mercaptoethanol (β-ME)) that contained the protease inhibitor cocktail tablet (Roche). After sonication, the lysate was cleared by centrifugation at 18,000 r.p.m. for 40 min at +4 °C. The soluble fraction was loaded onto the HisTrap FF Crude column (GE Healthcare). The non-specifically bound protein was removed with 10 and 60 mM imidazole, and the recombinant eEFSec was eluted with 300 mM imidazole. The eluate was diluted 3-fold in 20 mM HEPES pH 7.5, 50 mM NaCl, 0.5 mM TCEP and loaded onto the HiTrap SP HP ion-exchange column (GE Healthcare). eEFSec was eluted with a linear gradient of NaCl (0.1–1 M) in 20 mM HEPES pH 7.5, 0.5 mM TCEP. Fractions containing eEFSec were pooled and further purified on the size-exclusion chromatography column HiLoad 16/600 Superdex 200 (GE Healthcare) in 20 mM HEPES pH 7.5, 150 mM NaCl, 0.5 mM TCEP, 5 mM MgCl$_2$ (or 5 mM MnCl$_2$ for eEFSec:GDPNP). Pure eEFSec was concentrated to ∼8 mg ml$^{-1}$, flash-frozen in liquid nitrogen, and stored at −80 °C. The selenomethionine (SeMet)-derivatized human eEFSec (SeMet-eEFSec) was expressed by metabolic inhibition method and purified using the same protocol as for the native eEFSec. The only difference was that higher concentration of reducing agents was used in buffers (that is, 10 mM β-ME and 5 mM TCEP). A complete incorporation of 16 selenium (Se) atoms was confirmed by the mass spectrometry analysis.

**Site-directed mutagenesis.** The variant eEFSec constructs were prepared using QuikChange Site-Directed Mutagenesis Kit (Agilent Technologies) and mutations were confirmed by DNA sequencing. The variants were expressed in Lemo21(DE3) cells and purified using the Ni$^{2+}$-affinity and size-exclusion columns. The eEFSec variants were stored in 50 mM HEPES pH 7.5, 250 mM NaCl, 0.5 mM TCEP and 5 mM MgCl$_2$.

**Crystallization and data collection.** eEFSec was mixed with 1 mM of GTP analogue (GDPNP, GDPCP) or GDP. Equal volumes of the protein sample and reservoir solution were mixed and the crystals were grown at +12 °C using the sitting-drop vapor-diffusion method. The crystals of eEFSec:GDPNP:Mn$^{2+}$ were grown in 0.1 M HEPES pH 7.6, 0.15 M ammonium sulphate, 18% (w/v) PEG 3,350 and 0.02 M glycine. The same buffer supplemented with 0.25 M ammonium sulphate and 2% dextran sulphate supported growth of the eEFSec:GDPCP:Mg$^{2+}$ crystals. Because attempts to solve the crystal structure by molecular replacement were unsuccessful, the single-wavelength anomalous dispersion (SAD) phasing method based on SeMet was pursued. The crystals of SeMet-eEFSec complexed with GDPNP and Mn$^{2+}$ were grown against 0.1 M HEPES pH 7.6, 0.3 M ammonium sulphate, 16% (w/v) PEG 3,350, 0.02 M glycine and 4% dextran sulphate. By contrast, the crystals of eEFSec:GDP were grown in 0.1 M Tris pH 8.5, 0.2 M NaI, 0.2 M KI and 18% (w/v) PEG 3,350. The eEFSec:GDPN(C)P and eEFSec:GDP crystals were cryoprotected in mother liquor supplemented with 30% (w/v) PEG 3,350 and 18% (w/v) ethylene glycol, respectively, and flash-frozen in liquid nitrogen. The X-ray diffraction data were collected at liquid nitrogen temperature (λ = 0.97856 Å) at the Life Sciences Collaborative Access Team (LS-CAT) and the Structural Biology Center (SBC-CAT) beamlines of the Advanced Photon Source, Argonne National Laboratory (APS-ANL, Darien, IL), and scaled and reduced in HKL 3000 (ref. 53).

**Structure determination and refinement.** The crystal structure of eEFSec:GDPNP was determined by SAD phasing based on SeMet. Positions of selenium atoms were determined and the initial estimate of the experimental phase was calculated at 3.4-Å resolution in SHELX (ref. 54). FOM after SHELXD was 0.61 at 3.39 Å. After DM, R$_{cullis,ano}$ and FOM were 0.75 and 0.84, respectively. Density modification was done in DM[55] and an autobuild module of HKL3000 was used to trace the backbone in the experimental electron density map. Iterative model building was done in Coot[56] and structure refinement was done in Phenix[57]. The crystal structures of eEFSec:GDPCP and eEFSec:GDP were determined by molecular replacement using the structure of SeMet-eEFSec:GDPNP as a search model and Phaser[58]. In case of the GDP-bound structure, the molecular replacement solution was identified only after the search model was divided into the EF-Tu-like domain and the C-terminal D4. The final models of eEFSec:GDPNP, eEFSec:GDPCP and eEFSec:GDP refined to R$_{work}$/R$_{free}$ of 0.24/0.29, 0.24/0.29 and 0.30/0.34, respectively (Table 1), and they were of excellent geometry. The Ramachandran plots show that 84, 91 and 82% of residues of eEFSec:GDPNP, eEFSec:GDPCP, and eEFSec:GDP, respectively, are in preferred regions. Also, 16% (eEFSec:GDPNP), 9% (eEFSec:GDPCP) and 18% (eEFSec-GDP) of residues are in allowed regions. All figures showing the crystal structures of eEFSec and its complexes with guanine nucleotides were generated in PyMol[59]. The protein domain motion analysis was performed using DynDom online tool[60].

**Isothermal titration calorimetry.** The binding of guanine nucleotides (for example, GDP, GTP, GDPNP, GDPCP and GTPγS) to both the WT and mutant eEFSec constructs was monitored on the MicroCal ITC200 instrument. The binding events were measured in 50 mM HEPES pH 7.5, 250 mM NaCl, 0.5 mM TCEP and 5 mM MgCl$_2$, and the same buffer was used for the final eEFSec purification step and to dissolve the nucleotides. eEFSec (40–50 μM) was placed in

the sample cell and then titrated with the nucleotide solution (400–500 µM) while stirring. Heat changes resulting from productive binding were measured, integrated, and binding and thermodynamic parameters were calculated using MicroCal Origin software.

**In vitro activity assay.** The Sec incorporation activity of recombinant WT and eEFSec variants was determined using an in vitro translation system with a luciferase reporter mRNA, containing an in-frame UGA codon at position 258 followed by rat Gpx4 SECIS element in the 3′ UTR. Each 12.5 µl reaction contained 6.25 µl wheat germ extract, 320 nM recombinant SBP2, 320 nM wild-type or mutant His-tagged eEFSec, 20 µM amino acid mix, 125 ng of luciferase mRNA and 1.25 µg total aminoacyl-tRNA pool from rat testes (a rich source of selenium). The reactions were incubated at 25 °C for 2 h, and then luminescence intensity was measured using a 96-well plate luminometer.

**Small-angle X-ray scattering.** Samples containing eEFSec:GDP or eEF-Sec:GDPCP were at a final concentration of $\sim 5.4\,\mathrm{mg\,ml}^{-1}$ in 20 mM HEPES pH 7.5, 150 mM NaCl, 0.5 mM TCEP, 5 mM $MgCl_2$ and 1 mM nucleotide. SAXS experiments were conducted at the 18-ID Biophysics Collaborative Access Team beam-line (BioCAT), Advanced Photon Source, Argonne National Laboratory (APS-ANL), Chicago, IL[61]. Samples were exposed to X-rays using an in-line setup in which a 24 ml S200 column was directly coupled to the SAXS cell. Measurements taken before and after peak elution were used to establish the baseline scattering. BioCAT beamline specific pipelines, which use ATSAS suite[62] were used for data reduction. The Guinier Analysis and calculation of the radius of gyration, Rg, were done in PRIMUS[63]. Rg and the pair-distance distribution function, P(r), were calculated from the entire scattering pattern using GNOM[64], the low-resolution ab initio models were calculated in DAMMIF[65], and model clustering and averaging was done in DAMCLUST[66]. SUPCOMB[67] was used to superimpose the SAXS ab initio models onto the X-ray crystal structure. Finally, theoretical SAXS curves derived from the crystal structure were generated and overlaid with the experimental data in CRYSOL[68].

**Data availability.** All relevant data are available from the authors. Coordinates and structure factors described in this work have been deposited in Protein Data Bank under accession codes 5IZK (eEFSec:GDP), 5IZL (eEFSec:GDPCP) and 5IZM (eEFSec:GDPNP).

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

## Acknowledgements

We thank the staff of LS-CAT (21-ID) and SBC-CAT (19-ID) beamlines at the Advanced Photon Source (APS) of the Argonne National Laboratory (ANL) and organizers of the 7th Annual CCP4 USA Crystallography School for their help during X-ray data collection and processing, the staff at the Advanced Protein Characterization Facility (SBC-CAT, APS) for access to Mosquito crystallization robots and the staff of BioCAT (18-ID) beamline for help during SAXS data collection. We are grateful to Yury Polikanov for comments, suggestions and critical reading of the manuscript. The initial part of the study was supported by the University of Illinois at Chicago startup fund and a grant from the American Cancer Society, Illinois Division (225752 to M.S.). The subsequent studies were supported by grants from the National Institute of General Medical Sciences (GM097042 to M.S., GM070773 to P.R.C. and GM22854 to D.S.). This research used resources of the APS, a U.S. Department of Energy (DOE) Office of Science User Facility operated for the DOE Office of Science by ANL under contract no. DE-AC02-06CH11357. Use of the LS-CAT Sector 21 was supported by the Michigan Economic Development Corporation and the Michigan Technology Tri-Corridor (grant 085P1000817). BioCAT is supported by grant from the National Institute of General Medical Sciences of the National Institutes of Health (P41 GM103622). Use of the Pilatus 3 1M detector was provided by grant 1S10D018090-01 from the National Institute of General Medical Sciences.

## Author contributions

M.D.B. expressed, purified, characterized and crystallized eEFSec. M.D.B. and M.S. collected X-ray diffraction data, built, refined and analysed the crystal structures. M.D.B. and Z.O. calculated the experimental phase. M.D.B. prepared SAXS samples, S.C. collected SAXS data and calculated envelopes and M.D.B. and M.S. fitted structures into envelopes. M.H.P. and P.R.C. designed and completed *in vitro* activity assays. M.S. and P.R.C. designed the experiments. M.D.B., P.R.C., D.S. and M.S. wrote the manuscript.

## Additional information

**Competing financial interests:** The authors declare no competing financial interests.

