## [Peer Review File · Nature Communications]

Reviewers' Comments:

Reviewer #1 (Remarks to the Author)

In the manuscript by Dobosz-Bartoszek and colleagues, the authors present three crystal structures of human elongation factor eEFSec-responsible for delivering the essential amino acid selenocysteine to the ribosome in mRNA decoding of specific opal (UGA) stop codons-in the presence of GDP or two GTP analogs. The authors also use SAXS, ITC and mutagenesis to test their models for how eEFSec conformational changes they observe relate to its biological function. The authors find that the GDP structure compared to the GTP analog structures results in a 26-degree rotation of domain 4 (D4) relative to the rest of the protein. This structural rearrangement was confirmed using SAXS. The authors also find a rearrangement in positioning of D1 relative to D2, opening the GTP binding pocket and constricting the Sec binding pocket.

The results overall have the potential to be of interest for Nature Communications, but I found many parts hard to follow in its present form. These issues need to be addressed so that a wider audience can appreciate the work.

First and foremost, it was very difficult to ascertain how eEFSec relates to the myriad of GTPases that act on the ribosome. This makes much of the description hard to follow throughout. The authors very early on need to be clear about the distinctions between eEFSec and archaeal vs. bacterial SelB. Further, the authors need to be clear about the relationship to IF2/eIF5B much earlier in the manuscript. I suggest moving much of the comparison on pp. 10-12 to an earlier location, i.e. in the overall description of the structures, p. 5.

The authors also do not really show how the domain dynamics in eEFSec compare to IF2/eIF5B (p. 4, citing ref. 35, and top of p. 7). The authors should show what they mean by this. Supplementary Figure 6 doesn't capture this.

The authors also don't clearly lay out how their study addresses results presented in structural work on the archaeal SelB (ref. 29), and subsequent work questioning these results (ref. 31). This discrepancy is brought up initially on p. 6: "...but the results were questionable." How so? Do the authors really mean, "...but the results were questioned"? Why? It's not clear.

On p. 7 near the bottom, the comparison of the SAXS data to the crystal structures comes across in a confusing way. In particular, the sentence, "We conclude that eEFSec:GDP and eEFSec:GDPCP adopt the same structure in solution and in the crystal..." implies that there is just one conformation in all four cases. This is clearly not what the authors mean. Perhaps better would be to state, "We conclude that the conformational change between GDP and GTP states observed in eEFSec in crystals also occurs in solution. Thus the conformational change in the GDP-bound state of eEFSec is not a crystallization artefact."

In the GTPase center section at the bottom of p. 8, the authors state that EF-Tu and EF-G adopt the active conformation only when bound to the ribosome. The authors should be careful here. One of the primary reasons for the concern in ref. 31 is the fact that translation GTPases seem to have differential responses to different GTP analogs. For example GDPNP does not induce a correct positioning of the putative catalytic histidine in RF3 in Zhou et al., 2012, RNA 18, 230-ff, even with the factor bound to the ribosome. GDPNP also fails to induce proper positioning of the putative catalytic histidine in EF-G bound to the ribosome in Zhou et al., 2013, Science 340, 1236086. By contrast, GMPPCP allows correct positioning of the histidine (refs. 41-42; Pulk & Cate, 2013, Science 340, 1235970; Chen et al., 2013, NSMB 20, 1077-ff).

Related to the above discussion of GTP analogues, on p. 10 near the bottom, the authors have a

discrepancy in the text versus Table 2. From Table 2, both GDPNP and GDPCP have similar binding affinities to eEFSec as that observed for GTP. By contrast, GDP and GTP-gamma-S have similar affinities. I think the authors could eliminate the phrase about "the presence of GDP impurities" and keep the note on spontaneous hydrolysis of GTP-gamma-S. More importantly, the authors would be better served contrasting these affinities with those of EF-Tu for GDP and GTP off of the ribosome, which differ by a thousand-fold (could cite many different studies from the Rodnina lab).

Finally, the proposed mechanism of eEFSec function in decoding could be bolstered by more comparisons to eIF5B structures on the ribosome (Fernandez et al., 2013, Science 342, 1240585).

Reviewer #2 (Remarks to the Author)

The manuscript submitted by Dobosz-Bartoszek et al presents the structure of human selenocysteine tRNA-specific elongation factor (eEFSec) in complex with GDP as well as with GTP analogs. They also present the results of a directed structure-function analysis in order to determine the mechanistic importance of GTP hydrolysis and of the putative selenocysteine recognition site. The results reveal interesting mechanistic differences to canonical elongation factors (EF-Tu and EF1A), notably the absence of rotation of the D1 domain, and a novel conformational change of the D4 domain. Based on these results they propose a model for the incorporation of selenocysteine into proteins in eukaryotic cells that is distinct from the incorporation of canonical amino acids. The resented work is novel and of high quality. The writing does need some attention though and some minor comments follow below.

1. page 3: " ... and mutations in enzymes facilitating selenoprotein synthesis ... " Change mutations to changes, genes can be "mutated", but not enzymes or amino acids.
2. page 3: better formulation: "... this implies THAT the accurate decoding of the Sec codon ..."
3. page 3: better formulation: "Intriguingly, all aa-tRNAs ARE recognized and delivered ..."
4. page 4: Strange sentence, reformulate along the lines of variations of the functional and structural conservation "Although participating in the same process, the functional and structural conservation of eEFSec and SelB is not strict."
5. page 4: better formulation: "These observations raised THE question whether eEFSec, and by analogy SelB, promotes Sec incorporation by a mechanism distinct from the canonical mechanism based on EF-Tu."
6. page 5: "With the exception of flexible loops (residues 32-42, 70-80, 192-195, 383-403, 435-438, 524-526, and 544-569), the entire protein backbone was traced (Supplementary Fig. 1)." - supplementary Fig. 1 shows the GDP/GTP binding site, not sure to understand the link to the text.
7. page 5: "The N-terminal D1-3 folds into an EF-Tu-like structure (Supplementary Information) that harbors both the GTPase site and the putative Sec-binding pocket. The 6-stranded β -sheet of D1 is enclosed by 7 α -helices." Specify where exactly this information is displayed in the supplementary section.
8. page 6: . "The study on the archaeal SelB suggested otherwise 29, but the results were questionable 31." Not very elegant formation, specify what was "questionable" and find a more scientific way to express that the interpretation of past results pas possibly ambiguous.
9. page 7: "while the predicted 'canonical' model of eEFSec:GDP could not be superimposed onto the eEFSec:GDP envelope". Specify what the canonical model is.

10. page 8: "We found that although T48A, D92A, and H96A mutants bind GTP and GDP with high affinity." Change mutations to variants throughout the text, genes can be "mutated", but not enzymes or amino acids.

We sincerely appreciate comments and suggestions by reviewers. Below is our detailed response.

Reviewer #1:

- First and foremost, it was very difficult to ascertain how eEFSec relates to the myriad of GTPases that act on the ribosome. This makes much of the description hard to follow throughout. The authors very early on need to be clear about the distinctions between eEFSec and archaeal vs. bacterial SelB. Further, the authors need to be clear about the relationship to IF2/eIF5B much earlier in the manuscript. I suggest moving much of the comparison on pp. 10-12 to an earlier location, i.e. in the overall description of the structures, p. 5.

We concur with reviewer's suggestion and this discussion is placed immediately after description of the overall structure (see p. 6 and top of p. 7).

- "The authors also do not really show how the domain dynamics in eEFSec compare to IF2/eIF5B (p. 4, citing ref. 35, and top of p. 7). The authors should show what they mean by this. Supplementary Figure 6 doesn't capture this."

This is now clearly presented in Supplementary Figure 2d.

- "The authors also don't clearly lay out how their study addresses results presented in structural work on the archaeal SelB (ref. 29), and subsequent work questioning these results (ref. 31). This discrepancy is brought up initially on p. 6: "...but the results were questionable." How so? Do the authors really mean, "...but the results were questioned"? Why? It's not clear."

We rephrased the original statement on bottom of p. 7: "*The study on the archaeal SelB suggested otherwise, but the results were questioned because the functional states were captured by ligand soaking and not by co-crystallization.*"

- On p. 7 near the bottom, the comparison of the SAXS data to the crystal structures comes across in a confusing way. In particular, the sentence, "We conclude that eEFSec:GDP and eEFSec:GDPCP adopt the same structure in solution and in the crystal..." implies that there is just one conformation in all four cases. This is clearly not what the authors mean. Perhaps better would be to state, "We conclude that the conformational change between GDP and GTP states observed in eEFSec in crystals also occurs in solution. Thus the conformational change in the GDP-bound state of eEFSec is not a crystallization artifact."

We agree with the suggestion and the statement is now rephrased (see p. 9): "*We conclude that the conformational change between GDP and GTP states observed in eEFSec in crystals also occurs in solution. Thus, the conformational change in the GDP-bound state of eEFSec is not a crystallization artifact.*"

Also, we rephrased the title of Supplementary Figure 5 so that our point is more clearly explained.

- In the GTPase center section at the bottom of p. 8, the authors state that EF-Tu and EF-G adopt the active conformation only when bound to the ribosome. The authors

should be careful here. One of the primary reasons for the concern in ref. 31 is the fact that translation GTPases seem to have differential responses to different GTP analogs. For example GDPNP does not induce a correct positioning of the putative catalytic histidine in RF3 in Zhou et al., 2012, RNA 18, 230-ff, even with the factor bound to the ribosome. GDPNP also fails to induce proper positioning of the putative catalytic histidine in EF-G bound to the ribosome in Zhou et al., 2013, Science 340, 1236086. By contrast, GMPPCP allows correct positioning of the histidine (refs. 41-42; Pulk & Cate, 2013, Science 340, 1235970; Chen et al., 2013, NSMB 20, 1077-ff).

- Related to the above discussion of GTP analogues, on p. 10 near the bottom, the authors have a discrepancy in the text versus Table 2. From Table 2, both GDPNP and GDPCP have similar binding affinities to eEFSec as that observed for GTP. By contrast, GDP and GTP-gamma-S have similar affinities. I think the authors could eliminate the phrase about "the presence of GDP impurities" and keep the note on spontaneous hydrolysis of GTP-gamma-S. More importantly, the authors would be better served contrasting these affinities with those of EF-Tu for GDP and GTP off of the ribosome, which differ by a thousand-fold (could cite many different studies from the Rodnina lab).

We appreciate reviewer's input. These two points are now addressed on p. 12: "Lastly, it is important to mention significant corollaries derived from our studies. The question was raised whether GDPNP is a faithful mimic of GTP when bound to Sec elongation factors³¹. Our results unambiguously show that the GTPase site of eEFSec is the same when bound to GDPCP and GDPNP, and that both analogs trap eEFSec in the GTP-bound state (Fig. 5 and Supplementary Fig. 7). Consistent with our structural results, eEFSec binds GTP, GDPCP and GDPNP with similar affinities, while the binding affinity for GTP γ S is similar to that observed for GDP (Table 2 and Supplementary Fig. 6); the discrepancy among analogs is likely due to spontaneous hydrolysis of GTP γ S. In addition, unlike EF-Tu⁴⁵, eEFSec cannot discriminate GDP and GTP off of the ribosome (Table 2). Taken together, our data suggest that GDPNP and GDPCP are equally good analogs of GTP when bound to eEFSec and by extension to SelB, and that previous structural results on the archaeal SelB:GDPNP are valid²⁹. However, translational GTPases seem to have different responses to GTP analogs when bound to the ribosome. Namely, GDPNP failed to induce proper positioning of the catalytic His in RF3⁴⁶ and EF-G⁴⁷. By contrast, GDPCP allowed EF-Tu and EF-G to adopt the active conformation on the ribosome^{16,43,44,48,49}. Thus, it remains to be seen if both GTP analogs allow eEFSec and SelB to adopt the active conformation when in complex with the ribosome."

- Finally, the proposed mechanism of eEFSec function in decoding could be bolstered by more comparisons to eIF5B structures on the ribosome (Fernandez et al., 2013, Science 342, 1240585).

We extended our discussion on p. 13: "Moreover, it is plausible that D4 in eEFSec/SelB undergoes an additional rearrangement once bound to the ribosome and the A-site Sec-tRNA^{Sec}, analogous to IF2/eIF5B (Supplementary Fig. 2d). Namely, D4 of IF2/eIF5B translates ~4 Å upon nucleotide exchange, but then

***undergoes, together with D3 and linker, a much larger rotation (~50°) after interacting with the acceptor arm of the initiator tRNA and the ribosome (Supplementary Fig. 2d). The large domain movement and distortion of the tRNA body are of functional importance*³⁹.**

Reviewer #2:

1. page 3: "... and mutations in enzymes facilitating selenoprotein synthesis ..." Change mutations to changes, genes can be "mutated", but not enzymes or amino acids.

This is corrected as suggested.

2. page 3: better formulation: "... this implies THAT the accurate decoding of the Sec codon ..."

The sentence is rephrased as suggested.

3. page 3: better formulation: "Intriguingly, all aa-tRNAs ARE recognized and delivered ..."

The sentence is rephrased as suggested.

4. page 4: Strange sentence, reformulate along the lines of variations of the functional and structural conservation "Although participating in the same process, the functional and structural conservation of eEFSec and SelB is not strict."

The sentence is simplified: *"The decoding process exhibits structural and functional variations."*

5. page 4: better formulation: "These observations raised THE question whether eEFSec, and by analogy SelB, promotes Sec incorporation by a mechanism distinct from the canonical mechanism based on EF-Tu."

The sentence is altered as suggested.

6. page 5: "With the exception of flexible loops (residues 32-42, 70-80, 192-195, 383-403, 435-438, 524-526, and 544-569), the entire protein backbone was traced (Supplementary Fig. 1)." - supplementary Fig. 1 shows the GDP/GTP binding site, not sure to understand the link to the text.

The link is corrected to Figure 1 and Supplementary Fig. 1.

7. page 5: "The N-terminal D1-3 folds into an EF-Tu-like structure (Supplementary Information) that harbors both the GTPase site and the putative Sec-binding pocket. The 6-stranded β -sheet of D1 is enclosed by 7 α -helices." Specify where exactly this information is displayed in the supplementary section.

This is now specified in the main text, p 6: *"The N-terminal domain of human eEFSec resembles EF-Tu. Several structural differences that could be of functional significance have been noted. The overlay of D1 (r.m.s.d. of 1.5 Å*

shows that EF-Tu harbors two well-ordered α -helical insertions that sit atop the GTPase site. By contrast, these regions are shorter in eEFSec (residues 32-42 and 186-202) and partially disordered in our crystals. Likewise, the dorsal side of eEFSec harbors a partially disordered insertion (residues 57-87) where EF-Tu contains a well-ordered loop. Further, D2 from eEFSec and EF-Tu are similar (r.m.s.d. of 1.5 Å) with the only difference present in loop β 10- β 11, which is significantly shorter in eEFSec. Lastly, the overlay of D3 (r.m.s.d. of 1.7 Å) revealed that eEFSec contains insertions in several solvent-exposed loops: loop β 17- β 18 (residues 352-373), located at the dorsal face of eEFSec, β 21- β 22 (residues 432-444) at the interface of D1 and D3, and β 18- β 19 (residues 378-410)."

8. page 6: . "The study on the archaeal SelB suggested otherwise 29, but the results were questionable 31." Not very elegant formation, specify what was "questionable" and find a more scientific way to express that the interpretation of past results as possibly ambiguous.

We rephrased the original statement on bottom of p. 7: “*The study on the archaeal SelB suggested otherwise, but the results were questioned because the functional states were captured by ligand soaking and not by co-crystallization.*”

9. page 7: "while the predicted 'canonical' model of eEFSec:GDP could not be superimposed onto the eEFSec:GDP envelope". Specify what the canonical model is.

We defined the “canonical” model on p. 8: “*We wondered whether a reasonable structural model of the ‘canonical’ GDP-bound state of eEFSec, in which D1 adopts the same orientation as in the GDP-bound EF-Tu, could be designed.*”

10. page 8: "We found that although T48A, D92A, and H96A mutants bind GTP and GDP with high affinity." Change mutations to variants throughout the text, genes can be "mutated", but not enzymes or amino acids.

We concur with the suggestion and the text is altered as recommended. Also, mutants were replaced with variants throughout the text.

In addition to all these changes, we provided an updated Supplementary Figure 5 That contains Guinier and P(R) plots calculated from the SAXS data. These plots confirm the quality of the molecular envelopes.

We sincerely appreciate additional comments and suggestions by the reviewer. Below is our detailed response.

Major points:

On p. 7, line 10, the authors state that the ‘hinge’ region is a significant structural, but not functional element. I’m confused by this, as much of the later discussion is about movement of D4. The authors could state that K582A variant behaviour suggests that the salt bridge is structurally important, but additional experiments will be needed to assess its functional role.

The ⁵⁸²KRYVF⁵⁸⁶ -> AAAAA variant expresses poorly (suggesting misfolding), while the ⁵⁸³RY⁵⁸⁴ -> AA variant behaves just like the WT eEFSec. This led us to speculate that the “hinge” region may be of structural importance. However, we concur with the reviewer and rephrased the sentence on page 7:

“Our results suggest that the salt bridge between Gly372 and Lys582 is structurally important, but additional experiments are needed to assess its functional role.”

What happens in the GDP vs. GDPCP structures? Is the salt bridge retained?

The salt bridge is retained in both the GDP and GDPCP structures.

Also please show the location of the salt bridge in both states in Supplementary Figure 3, and possibly in the Supplementary Movie.

The salt bridge in both structures is shown in Supplementary Figure 3b. The functional data are now moved to Supplementary Figure 3c.

Minor points:

1. On p. 3 line 9, the use of “mutations” is justified here, as these were identified in animals.

Corrected as suggested.

2. On p. 4 line 6 new sentence, presumably “The Sec decoding process”?

Corrected as suggested.

3. On p. 6, first paragraph, cite ref. 29 here.

Corrected as suggested.

4. On p. 6, last two sentences. Perhaps a better wording:

Although conservation of the archaeal and human D4 is not strict, as the human enzyme harbors an additional α -helix and a longer C-terminal segment (Supplementary Fig. 2a), closer inspection...

Corrected as suggested.

5. On p. 7 line 1, “an H-bond”

Corrected as suggested.

6. Middle of p. 9, should be “artefact”, not “artifact”.

“Artifact” is the American spelling of the British “artefact”. We leave to the editorial staff to decide what version to be used in the manuscript.

7. On p. 9, 5 lines from bottom of 1st paragraph, should note this is an earlier finding by Ban and co-workers with archaeal SelB.

Corrected as suggested.

8. On p. 16, provide citation that rat testes are enriched in Sec-tRNA(Sec).

After careful review of the pertinent literature, we conclude that the original statement is misleading. Namely, it is well established that selenium (and selenoprotein) levels are particularly high in male reproductive organs in mammals (see: Brown DG and Burk RF, J. Nutr., 1973 and Behne D et al., J. Nutr., 1982, among others). Because most of selenium is used for selenoprotein synthesis, one could assume that levels of Sec-tRNA^{Sec} are higher in testes. However, this was not measured and we altered the statement into “a rich source of selenium”.

9. On p. 16-17, please provide citations for all of the SAXS software (developers need citations to enable future funding).

References 61-68 are provided.

10. In figure legend for Supplementary Fig. 4, please define species abbreviations for the general reader.

Completed as suggested.

11. The graphical abstract is not very helpful in conveying the conformational dynamics in the D4 region. Could use arcs like those commonly used in cartoons to denote motion.

Arrows are replaced with arcs in Figure 6 as suggested.